# Vaccination against Tick-Borne Encephalitis (TBE) in Italy: Still a Long Way to Go

**DOI:** 10.3390/microorganisms10020464

**Published:** 2022-02-18

**Authors:** Donatella Panatto, Alexander Domnich, Daniela Amicizia, Paolo Reggio, Raffaella Iantomasi

**Affiliations:** 1Department of Health Sciences, University of Genoa, Via A. Pastore 1, 16132 Genoa, Italy; panatto@unige.it; 2Hygiene Unit, San Martino Policlinico Hospital, Largo R. Benzi 10, 16132 Genoa, Italy; alexander.domnich@hsanmartino.it; 3Vaccine Medical Department, Pfizer Srl., Via Valbondione, 113, 00188 Roma, Italy; paolo.reggio@pfizer.com (P.R.); raffaella.iantomasi@pfizer.com (R.I.)

**Keywords:** tick-borne encephalitis, TBE, tick-borne encephalitis virus, vaccination, TBE vaccine, vaccination strategies, cost-effectiveness assessment

## Abstract

Tick-borne encephalitis (TBE) is endemic in several European countries, and its incidence has recently increased. Various factors may explain this phenomenon: social factors (changes in human behavior, duration and type of leisure activities and increased tourism in European high-risk areas), ecological factors (e.g., effects of climate change on the tick population and reservoir animals), and technological factors (improved diagnostics, increased medical awareness). Furthermore, the real burden of TBE is not completely known, as the performance of surveillance systems is suboptimal and cases of disease are under-reported in several areas. Given the potentially severe clinical course of the disease, the absence of any antiviral therapy, and the impossibility of interrupting the transmission of the virus in nature, vaccination is the mainstay of prevention and control. TBE vaccines are effective (protective effect of approximately 95% after completion of the basic vaccination—three doses) and well tolerated. However, their uptake in endemic areas is suboptimal. In the main endemic countries where vaccination is included in the national/regional immunization program (with reimbursed vaccination programs), this decision was driven by a cost-effectiveness assessment (CEA), which is a helpful tool in the decision-making process. All CEA studies conducted have demonstrated the cost-effectiveness of TBE vaccination. Unfortunately, CEA is still lacking in many endemic countries, including Italy. In the future, it will be necessary to fill this gap in order to introduce an effective vaccination strategy in endemic areas. Finally, raising awareness of TBE, its consequences and the benefit of vaccination is critical in order to increase vaccination coverage and reduce the burden of the disease.

## 1. Introduction

Tick-borne encephalitis (TBE) is an acute infection of the central nervous system; it is caused by a virus, which is usually transmitted to humans by a tick bite [1]. Tick-borne encephalitis virus (TBEV) is a member of the genus flavivirus. It has a linear positive-stranded RNA genome that consists of a single open reading frame encoding for one polyprotein, which is the precursor of three structural proteins—E (envelope), C (capsid), and M (membrane)—and seven non-structural proteins [1]. The E protein is the main target of TBEV-neutralizing antibodies [2].

TBEV is usually divided into three subtypes: (1) European (TBEV-Eu); (2) Siberian (TBEV-Sib); and (3) Far Eastern (TBEV-FE). The main hosts of TBEV are small rodent species, deer, and sheep, while ticks (*Ixodes persulcatus* and *Ixodes ricinus*) serve as vectors. The vector of TBEV-Eu is *Ixodes ricinus*, while *Ixodes persulcatus* is the main vector of the other two subtypes [3]. *Ixodes ricinus* is found in most of Europe, and its range also extends to Turkey, northern Iran, and the southeastern Caucasus. *Ixodes persulcatus* is found in a belt that extends from Eastern Europe to China and Japan. Both tick species co-circulate in a restricted area in northeastern Europe, Russian Karelia, Saint Petersburg, and eastern Estonia and Latvia [4,5,6].

Viral transmission to a tick takes place when ticks at various developmental stages (particularly nymphs and larvae) co-feed on the same animal in specific climatic conditions (for instance, during rapid warming in springtime) [3].

The transmission of TBEV to humans occurs through the saliva of an infected tick within minutes of the tick bite [3]. Another important route of transmission is through the consumption of virus-infected, unpasteurized dairy products [7].

TBE follows a median incubation period of 8 days (range 4–28) [8,9,10]. The lead time to clinical symptoms is shorter (3–4 days) in the case of foodborne infections.

TBEV infections and the resulting disease display a first viremic phase, which can progress to a second (neurological) phase [11]. The virus spreads systemically during the first phase, producing fever, headache, fatigue, myalgia, anorexia, nausea, and/or vomiting [9,11]. In some patients, the virus does not invade the central nervous system (CNS) and the disease terminates after the first phase (monophasic disease). In others, the virus penetrates the CNS, causing a second phase of illness, with neurological signs and symptoms (biphasic TBE) [8,11]. In the second stage, clinical manifestations include mild meningitis or severe encephalitis, with or without myelitis, and spinal paralysis [8,9,11]. In about 5–10% of cases, monoparesis, paraparesis, and tetraparesis can develop [8]. Cranial nerve involvement is associated with ocular, facial, and pharyngeal motor dysfunction, as well as vestibular and hearing defects. In severe cases, brainstem involvement can lead to substantial respiratory and circulatory failure [8]. In some TBE patients, TBEV can remain active in the CNS for a long period, and the infectious process continues. In this case, TBE becomes a chronic (progressive) disease [11].

Given the relatively severe clinical course of TBE, the absence of any antiviral therapy, and the fact that many patients do not fully recover, this disease is an increasingly worrying public health problem [12].

The incidence of clinical cases of TBE is reported to be between 10,000 and 15,000 per year worldwide, though it is probably underestimated, as notification of the disease is not mandatory in all countries [1,13].

The aim of this overview was to describe the epidemiology of TBE in Italy in the European context and to identify possible instances of underestimation of the disease, in order to implement vaccination programs and propose viable strategies for reducing the burden of TBE in terms of clinical complications and healthcare expenditure. Furthermore, in order to assess the increasing public health concern associated with TBE as a whole, the risk of infection during travel to European endemic areas was assessed.

## 2. Epidemiology of TBE in Europe

TBE is endemic in areas from Alsace-Lorraine and Scandinavia to North-Eastern China and Northern Japan [14], and the last several decades have seen an increase both in areas of TBEV endemicity and in the total number of TBE cases reported.

Since 2012, TBE has been a notifiable disease in the European Union. The European Centre for Disease Prevention and Control (ECDC) annually collects data from 28 European countries (not included Russia and Switzerland).

Between 2015 and 2019, 23 countries reported 13,842 TBE cases, more than 40% of which were reported by two countries (Czech Republic and Lithuania); by contrast, Ireland and Spain reported 1 case each, and 3 countries reported no cases (Greece, Luxemburg and Romania). The notification rate was highest in Lithuania (25.4 cases per 100,000 population), followed by Czechia (7.3) and Estonia (6.2) [13].

Notably, an increase has been recorded in France, where the notification rate in the Alsace region almost tripled in 2016 compared with previous years [15]. Moreover, some traditionally non-endemic or TBE-free countries, such as Belgium and the Netherlands, have reported possible new endemic foci, with the Netherlands reporting its first locally acquired human case in 2016 [15].

In 2019, the EU/EEA notification rate was 0.7 per 100,000, representing a slight increase from the stable rate of 0.6 reported in the three previous years. The highest notification rates were reported in Lithuania, Czechia, and Estonia (Figure 1) [13].

Notification rates are higher among males and among adults aged 45–64 years, probably owing to their more frequent exposure to tick bites during outdoor activities associated with occupation or leisure [13].

In July 2019, the first report of a probable human case of TBE in the UK was published. In September 2019, TBEV was detected in ticks in southern England, which suggested the need to consider TBE as a diagnosis in encephalitis patients in the UK [13].

A substantial increase in TBE incidence has also been recorded in the Czech Republic, which currently has the highest number of cases in the European Union and has long been one of the countries with the highest incidence, together with the Baltic countries and Slovenia [16]. In 2019, a descriptive analysis was made in the Czech Republic; the epidemiological characteristics of TBE cases by place and time were described and compared with the previous 5-to-10-year period [17]. Specifically, in 2019, a total of 773 cases were reported; this represents an incidence of 7.24 cases per 100,000 inhabitants, constituting a 40.6% increase from the 2014–2018 average and the second highest number in the previous 10 years, with the highest incidence (8.20 cases per 100,000 inhabitants) being recorded in 2011 [17]. 

A feature of TBE is that the incidence of the disease in risk areas can vary significantly from year to year. In addition to short-term fluctuations, there are also longer-range undulations of incidence rates [18].

The expansion of endemic areas and increased incidences have been associated with various factors, including ecological changes that favor tick reproduction, increases in human outdoor activities, and climatic changes favoring virus circulation in natural foci. For example, in Austria, a study published in 2015 analyzed changes in the incidence of TBE in the unvaccinated population over a period of 42 years. The overall incidence in the country remained constant, but new endemic areas were detected in previously unaffected western Alpine regions, and the increased incidence in these areas contrasted with the decline in cases located in traditionally endemic eastern areas [18].

A recent paper has claimed that climate change plays only a minor role in the rising trend of Austrian TBE cases, and that population growth is the most probable cause of this trend. However, the authors state that this increase is probably a multifactorial phenomenon, with a complex interplay of factors that influence the dynamics of viral transmission in natural hosts, including climate change and human factors, such as risk exposure, better diagnostic capabilities, and population aging [18,19,20]. Furthermore, another study showed that other factors may explain the observed oscillation of the annual incidence of TBE in Austria. In their modelling study of a 40-year trend of TBE cases, they established that the main determinant of the variation in TBE incidence was the natural cycle of TBE transmission between rodents and ticks, which is driven by beech fructification. Population growth, net migration index, large-scale climate changes, and annual sunshine duration (as a proxy of human outdoor activity) may also contribute to the observed oscillations in annual TBE incidence [20].

## 3. Epidemiology of TBE in Italy

Italy is considered to be at low risk for TBE, which appears to be restricted to a relatively small fraction of the country, with cases of TBE being recorded in the north-eastern and central areas [21]. The map of Italy shown in Figure 2 reports the areas with the highest incidence.

The first cases of TBE in Italy were identified in Tuscany in the 1970s [22,23]. Other foci were subsequently reported in the 1990s in the northern provinces of Trento (Trentino Alto Adige) and Belluno (Veneto) [24,25], and new autochthonous cases were detected in the Friuli Venezia Giulia region a few years later [26].

The incidence of TBE in Italy has shown a rising trend since the early 2000s, from 12 cases reported in 2000 (0.2/100,000 inhabitants) to 55 cases in 2020 (1.42/100,000) (Figure 3) [27]. 

Riccò M et al. recently reported a total of 103 cases in Italy between 2017 and 2020, 100 of which were in the Triveneto area (i.e., autonomous provinces of Trento and Bolzano, and the Veneto and Friuli-Venezia-Giulia regions), with a pooled incidence rate of 0.35 per 100,000 [95%CI 0.28–0.42], highlighting the fact that TBEV endemicity is restricted to this part of the country [28].

In a cross-sectional study (2007–2018) performed to ascertain the burden of TBE in the Veneto region (north-eastern Italy), cases of TBE occurring from 1 January 2007 to 31 December 2018 were extracted from the database of the mandatory notification system (MNS) and from hospital discharge records (HDRs) [12]. Capture–recapture methods were used to estimate the completeness of each data source (as a percentage of cases) [12]. There were 192 cases of specific TBE according to the HDRs, while 244 cases were registered in the MNS. A total of 281 patients were specifically diagnosed with TBE; 55.2% (155 cases) were identified from both the HDRs and the MNS, while 31.7% (89 cases) were only registered in the MNS, and 13.2% (37 cases) only in the HDRs (Figure 4) [12].

**Figure 2 microorganisms-10-00464-f002:**
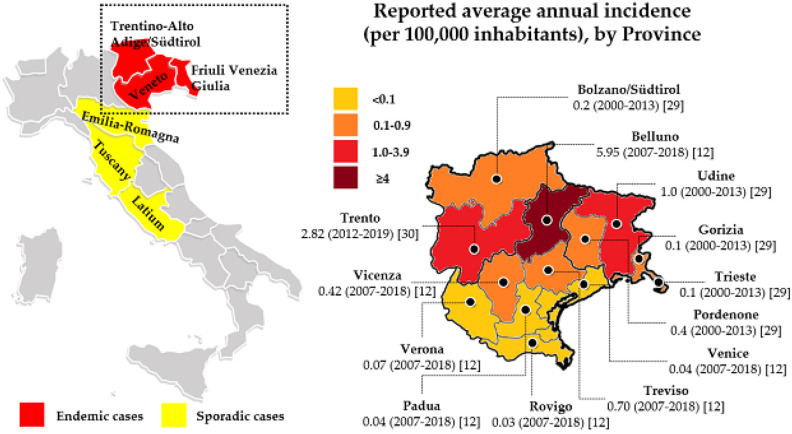
Map of the Italy showing areas where TBE is present. The average annual incidence in the north-eastern area is shown in detail [12,29,30].

**Figure 3 microorganisms-10-00464-f003:**
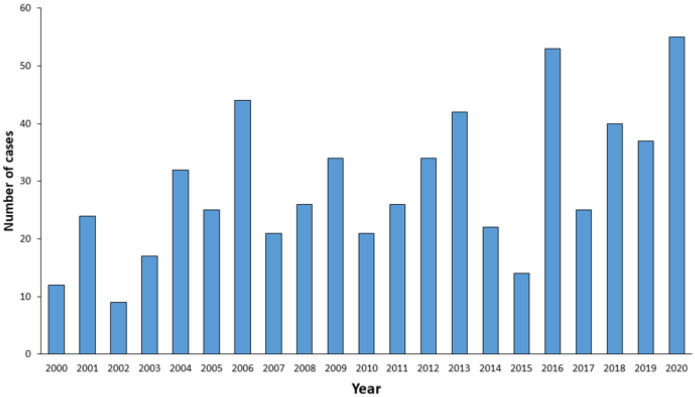
Annual number of TBE cases reported in Italy from 2000 to 2020 (Adapted from ref. [27]).

**Figure 4 microorganisms-10-00464-f004:**
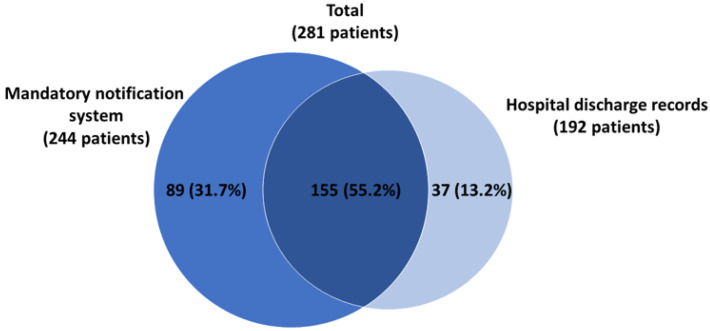
Venn diagram showing the number and percentage of cases of TBE identified by 2 sources: the MNS and HDRs in Veneto from 2007 to 2018. MNS, mandatory notification system; HDRs, hospital discharge records. (Adapted from ref. [12]).

Overall, the capture–recapture method used by these authors was able to identify a total of 302 cases of TBE in the Veneto region in the previous 12 years, indicating that pooling the data from the two sources considered (HDRs and MNS) could provide a fairly complete picture of the situation (93% of cases); by contrast, when considered separately, the two sources yielded lower values: 80.8% for the MNS (95% CI 77.6–84.3%) and 63.6% for the HDRs (95% CI 61.0–66.3%) [12].

In the Veneto region, the annual incidence of TBE observed during the study period (2015–2018) was 0.48 per 100,000 population, with high variability among areas: from 0.03 per 100,000 in the province of Rovigo to 5.95 per 100,000 in the province of Belluno [12]. This study identified a rise in the reported frequency of TBE in the 2015–2018 period, which was significant in mountainous areas and peaked in the last year considered (2018) (Figure 5) [12].

These findings confirmed those of a retrospective study conducted in an area of north-eastern Italy from 2000 to 2013, which showed that cases of TBE were increasing [29]. During this period, 367 cases (0.38/100,000 inhabitants) were diagnosed in the study area. Almost all subjects (*n* = 355) involved were Italian nationals and the majority (*n* = 364) also resided in the study area; most patients were males (*n* = 257; 70%) and about 70% (*n* = 255) were between 30 and 70 years old. Encephalitis was the most common clinical presentation (*n* = 175 cases), while febrile syndrome without CNS involvement (*n* = 60 cases) accounted for less than a sixth of the cases. Almost all patients (*n* = 364) was unvaccinated. Sequelae (tiredness, tremor, headache, memory and/or concentration disturbances, pain in the extremities, insomnia, anxiety, nausea, hearing disturbance, dizziness, or electroencephalography anomalies) were reported in 60 (16.3%) individuals. The case-fatality rate was 0.5% [29]. In 156 of the 367 cases, the setting of tick bite exposure was known: recreational activities (trekking, mushroom picking) were mentioned in 111 cases, living in a high-risk area was reported by 32 individuals, agricultural work was reported by 9 subjects, and travel abroad (Austria and Slovenia) was reported by 4. The total number and incidence rate (IR) of cases found in this study were much higher than those previously reported in the same geographical area [29]. 

The province of Trento (part of the Trentino-Alto Adige region), where a low incidence (0.6) of TBE was registered between 1992 and 2004, has recently been identified as one of the three provinces of Triveneto with the highest TBE incidence (IR 2.82). Indeed, a recent study was performed to identify active TBEV foci and to assess the current TBE hazards in this province by applying an integrated approach that combines the distribution of human cases, the seroprevalence of TBEV in sentinel hosts, and the screening of questing ticks for TBEV. The study revealed an increase in TBEV prevalence in ticks and the emergence of new active TBE foci, in comparison with previous surveys, and demonstrated the importance of an integrated approach to TBE risk assessment [30].

Since the observation of the first cases of TBE in the Friuli Venezia Giulia (FVG) region in 2004, several cases have been recorded (a total of 79 documented cases in 2015), 40% of which were in the Alto Friuli area [31].

Examination of the data from the Italian official surveillance system reveals that no notifications of TBE cases were registered in Lombardy (a region of northern Italy bordered by Switzerland), even though the vector *Ixodes ricinus* is present. Therefore, improved surveillance of tick-borne pathogens should include a search for etiological agents in all suspected cases [32].

Sporadic cases of TBE have also been detected outside the known endemic areas in Italy. For example, two autochthonous cases were reported in 2020 in the Emilia Romagna region [31].

## 4. Clinical Burden of TBE

Although the mortality rate in adult patients infected with a European strain of TBEV is <2% [33] and a recent European multicenter study reports a fatality rate of 0.9% [34], TBE may result in long-term or permanent neurological sequelae that can impair the patient’s quality of life and require significant lifestyle changes.

Four prospective studies [8,9,10,35] assessing the long-term morbidity associated with TBEV-Eu have reported the existence of a post-encephalitic syndrome, with 26–46% of patients presenting residual symptoms after 6–12 months, and 30% showing moderate-to-severe impairment of quality of life owing to neurological dysfunction [9].

In this regard, a study aimed to evaluate the clinical consequences of TBE disease and categorized sequelae into three main groups: minor complaints (10%), major complaints (46%), and difficulties associated with resuming previous levels of activity (2%). Mild sequelae (emotional lability, tiredness, and intermittent headache) did not have a significant impact on patients’ quality of life, while moderate sequelae (gait ataxia, paresis of extremities, marked dementia, or severe deafness) affected their quality of life, daily activities, and social and working capacity. In patients with severe sequelae, social life and working capacity were seriously affected [36]. The burden measured in disability-adjusted life years (DALYs) was estimated by the same research team by using the updated DALYs methodology, whereby DALYs are the sum of the number of life years lost owing to premature death and the number of life years lost because of disability, weighted with a factor between 0 (perfect health) and 1 (death) according to the severity of the disability [37]. The authors found a total number of 3450 DALYs from the population perspective and of 3.1 DALYs per case from the individual perspective, suggesting that TBE carries a heavy burden from both these perspectives and that neurological sequelae have the greatest impact on the overall TBE burden. Indeed, more than 90% of DALYs lost are due to long-term neurological disability in survivors [38].

Other researchers have investigated the long-term outcomes of post-encephalitic TBE syndrome, diagnosed in 35–58% of their patients. The most commonly reported symptoms were cognitive or neuropsychiatric complaints (reduced stress tolerance and impaired memorization ability), balance disorders, headache, dysphasia, hearing defects, and limb paresis [8,39,40].

## 5. TBE Vaccines Available in Europe

TBE can be prevented by active immunization. Four TBE vaccines are currently on the market: FSME-IMMUN/TICOVAC (Baxter; now produced and marketed by Pfizer), Encepur (Novartis vaccines, then GSK, and recently Bavarian Nordic), EnceVir (the Tomsk branch of the Federal State Unitary Enterprise “*Mikrogen*”, Russia), and another Russian vaccine from the Institute of Poliomyelitis and Viral Encephalitis (the IPVE vaccine, also called the TBE vaccine Moscow).

The two vaccines available in the European Union use TBEV-Eu strains:-FSME-IMMUN/TICOVAC (strain Neudoerfl). This is the only vaccine marketed in Italy.-ENCEPUR (strain K23).

The viral particles used in both vaccines are propagated in chick embryo cells, filtered and inactivated in formaldehyde, and further purified by ultracentrifugation. The whole inactivated virus is adsorbed to aluminum hydroxide and stabilized with human albumin (FSME-IMMUN/TICOVAC) or sucrose (ENCEPUR), and is free from thiomersal [41]. 

Conventional immunization schedules are similar for both vaccines, with two intramuscular doses administered 1–3 months apart and a third dose given before the next tick season. This schedule, in the case of both vaccines, induces antibodies that are deemed to be protective in over 90% of children and adults. Because of the gradual decline in antibodies after the third dose, a booster is needed after 3 years. After this fourth dose, a more stable antibody titer is maintained in most individuals, allowing a longer booster interval of 5 years [42].

Accelerated schedules for use during the endemic season have been introduced for FSME-IMMUN/TICOVAC, with a shortened interval of 2 weeks between the first two doses [43], and for ENCEPUR, with three doses being given on days 0, 7, and 21 [44].

## 6. Focus on FSME-IMMUN/TICOVAC Vaccine

As this overview focuses on the Italian context, only FSME-IMMUN/TICOVAC is evaluated.

Several studies aimed at evaluating the immunogenicity of FSME-IMMUN/TICOVAC in both adults and children on a conventional schedule have been carried out, and have shown good results [45,46,47,48,49,50].

The primary vaccination schedule consists of three doses. The first and second dose should be given at a 1-to-3-month interval. The third dose should be administered from a minimum of 5 to 12 months after the second dose. After receipt of the third dose, the primary series is considered “complete” [51].

However, vaccination with FSME-IMMUN/TIVOVAC provides sufficient protection for the upcoming tick season after just two vaccine doses. Indeed, as early as 14 days after the second dose, the immune system can generate sufficient levels of antibodies to protect the patient against TBEV infection, while a third dose is needed for longer protection [51,52,53]. 

Indeed, clinical studies performed to evaluate seroconversion rates after TBE vaccination have shown that the highest antibody levels are reached after administration of the third vaccine dose, suggesting that it is essential to complete the whole vaccination schedule in order to achieve protective antibody levels in most vaccinees [47]. Nevertheless, rapid immunization schedules, with an interval of 2 weeks between the first two doses, have been seen to elicit high antibody titers 14 days after administration of the second dose (89.3%), and 7 days after administration of the third dose (91.7%) [43].

The results of a follow-up study assessing sero-persistence after vaccination support the need to administer the first booster dose within 3 years after the primary vaccination. After the first booster dose, high levels of TBE antibodies were maintained for up to 5 years, both in subjects aged <50 years (94.3%) and in those aged 50–60 years (>90.2%), suggesting an optimal interval of 5 years between booster doses in all subjects aged <60 years [54]. Thus, it is currently recommended that subjects aged <60 years should receive further booster vaccinations every 5 years and those aged 60 years or more should receive further booster vaccinations at least every three years [51].

Clinical studies in the pediatric population (1–15 years of age) [50,55,56] have yielded similar results to those obtained in adults, supporting the use of an analogous vaccination schedule with a lower vaccine dose (1.2 μg) [49].

In addition to the clinical studies, the “Austrian experience” constitutes the first attempt to evaluate vaccine effectiveness. In the pre-vaccination period, Austria had the highest level of TBE morbidity in Europe, with more than 50% of all viral meningoencephalitides caused by TBEV infection [57]. After initiation of the mass vaccination campaign in 1981, the vaccination coverage of the population increased from 6% in 1980 to 86% in 2001, exceeding 90% in some high-risk areas, with a protection rate >90% after three doses of the vaccine [57]. This increased vaccination coverage led to a substantial decline in TBE, reducing the burden of the disease in Austria [57].

A later study assessed the effectiveness of TBE vaccination in Austria in the years 2000–2006 in several age groups by considering unvaccinated and vaccinated hospitalized patients. In subjects who had completed the course of vaccination, overall effectiveness was about 99%, with no statistically significant difference between age groups [58]. A further follow-up study, which used data on vaccination coverage in Austria and TBE incidence rates among non-vaccinated and vaccinated populations, calculated the effectiveness of vaccination in the period 2000–2011. Overall effectiveness among fully vaccinated persons was 99%, regardless of age (although antibody titers were lower in elderly patients) [18].

Pugh et al. [52] recently published the results of a study in which surveillance and vaccination data from 2018 to 2020 were analyzed in order to determine the clinical protection afforded by two or three or more doses of TBE vaccine. Vaccine effectiveness was consistently high across two-dose-only and ≥three-dose vaccination categories (94.6–97.4%). This analysis bolsters evidence from the clinical development program, showing a robust immune response and short-term sero-persistence following two doses of FSME-IMMUN/TICOVAC. These data support the strategy of administering two doses of the TBE vaccine to travelers who do not have enough time to complete the full three-dose primary series prior to departure.

The effectiveness of TBE vaccination was also evaluated in a study conducted in southern Germany and Latvia from 2007 to 2018. The authors reported that vaccine effectiveness after two, three, and ≥four doses was high in both countries, being 97.2%, 95.0%, and 95.4% in southern Germany, and 98.1%, 99.4% and 98.8% in Latvia, regardless of age [53].

The impact of age on vaccine efficacy is still controversial. Indeed, although the immune response is better in younger subjects, the results of several studies do not suggest that older people are at higher risk of vaccine breakthrough [59,60]. The sero-response rate induced by FSME-IMMUN/TICOVAC after the primary vaccination schedule in subjects 70–87 years of age has been reported to be 99.3%. 

An association between age and vaccine breakthroughs was observed in a Swedish study. However, it is noteworthy that approximately half of the patients in whom vaccine failure was observed had comorbidities affecting the immune system, suggesting that other factors could explain the higher proportion of vaccine failure in subjects aged ≥50 years [61]. However, there is no evidence that an additional priming dose is of “immunological” or clinical benefit [61,62].

Data on the minimum duration of protection and booster intervals of TBE vaccination are of great interest. Consequently, in the last decade, several studies have investigated long-term sero-persistence and intervals between boosters [63,64,65,66,67].

A long-term prospective follow-up study was the first to document antibody persistence up to 10 years after a primary vaccination course and a first booster with FSME-IMMUN/TICOVAC. The authors found that, after a complete primary series and one booster dose 3 years later, a high proportion of subjects in the groups aged 18–49 and 50–60 years remained seropositive up to 5 years. Furthermore, 10 years after the first booster, the seropositivity rate was seen to have decreased to 88.6%, 74.5%, and 37.5% in subjects aged 18–49 years, 50–60 years, and >60 years, respectively. These results confirmed the correctness of the current booster recommendations for subjects >60 years of age (i.e., 3-year booster interval) [65].

The induction of immunological memory, which is characterized by a rapid and sustained secondary immune response, is proving to be an alternative mechanism of protective action against TBE. In this context, Switzerland and Finland have adopted a longer booster interval (10 years) following the three-dose primary immunization schedule, with no evidence of harm at the population level [59]. Extending booster intervals would probably improve vaccine uptake, reduce costs, and improve acceptability [63].

## 7. Cost-Effectiveness Studies of TBE Vaccines

In addition to the clinical impact of TBE disease, it is important to underline its economic impact. Long-term or permanent neurological sequelae, in addition to hospitalization costs, determine high costs for healthcare systems, insurance companies, and society. In a study conducted in Slovenia, the weighted average cost of hospitalization (calculated from costs of hospitalization and the probabilities of patients who need hospitalization) was EUR 6457 per year for patients aged 18–60 years, and EUR 6876 for those aged >60 years, whereas the average cost associated with permanent severe neurological sequelae was EUR 28,952 per year [36]. Similarly, the economic burden of TBE in Russia in 2011 was estimated to be about USD 49.5 million, 78% of which was attributed to death and disability [68].

In order to evaluate the clinical and economic benefit of vaccination programs, several studies on the cost-effectiveness of TBE vaccination have been carried out. Specifically, the total economic benefits of vaccination campaigns in Austria between 1981 and 1990, in terms of reductions in inpatient care costs, lost productivity, and premature retirement, were evaluated at EUR 24 million [69]. Similarly, the estimated economic benefits of vaccination campaigns between 1991 and 2000 amounted to EUR 60 million.

Endemic areas where reimbursed vaccination programs have been introduced include Austria, Germany, Slovenia, Switzerland, parts of Finland, Hungary (for patients with an occupational risk), and Latvia (for children under 18 living in endemic areas). In many cases, the decision to introduce the program was based on a cost-effectiveness assessment (CEA) [70].

In 2012, a study on the cost-effectiveness of the two vaccines licensed in Western Europe was published; this showed that, in Slovenia, vaccinating adults aged 18–80 years was cost-effective from the perspective of the healthcare payer, and also cost-saving from the societal perspective. Specifically, the ICER for vaccination amounted to approximately EUR 15,000–20,000 per QALY gained from the perspective of the healthcare payer. On placing a limit of EUR 30,000 per QALY on willingness to pay in Slovenia, vaccination against TBE proved to be a cost-effective intervention. From the societal perspective, vaccination was also cost-saving, as it avoided the high indirect costs associated mainly with lost productivity [36].

In subsequent cost-effectiveness studies on TBE in Sweden, Finland and Estonia, TBE vaccination proved highly cost-effective [71]. Specifically, one such study performed in Finland showed that, from the healthcare payer’s perspective, the introduction of vaccination was cost-saving in areas where the incidence of TBE was around 15/100,000. Moreover, it remained cost-effective until the incidence declined to 5/100,000. On the basis of this cost-effectiveness study, the TBE immunization working group in Finland recommended vaccination against TBE [71].

An Estonian study showed that vaccinating the whole population against TBE would produce an ICER of EUR 61,000 per QALY gained. Vaccinating the population aged ≥50 years proved more cost-effective from the healthcare perspective, with an ICER of EUR 25,000 per QALY gained [71].

Moreover, a recent Swedish analysis of the cost-effectiveness of a publicly funded vs. an out-of-pocket TBE vaccination program in Stockholm found that a structured vaccination program was cost-effective, particularly for younger individuals [72]. The ICERs were 30,000, 100,000, and 160,000 SEK/QALY in cohorts aged 3, 40, and 50 years, respectively; these values are well below the estimated cost-effectiveness threshold in Sweden, which may be between 250,000 SEK (EUR 24,024) and 1,000,000 SEK (EUR 96,099), according to the severity of the disease (Sweden does not have an explicit willingness-to-pay limit). The cost-effectiveness acceptability curve showed that, at any cost-effectiveness threshold, the probability of vaccination being cost-effective was higher for the 3-year-old cohort than for the other groups [72]. On the basis of these results, vaccination against TBE can be considered a cost-effective intervention in the Swedish endemic area.

## 8. Vaccination against TBE in Italy

The Italian National Vaccination and Prevention Plan for 2017–2019 recommends vaccination for residents and at-risk workers (farmers or soldiers) in endemic areas and for those who visit rural or forest environments up to an altitude of 1400 m in these areas [42]. This plan was extended to 2021, owing to the COVID-19 pandemic. Furthermore, TBE vaccination is also recommended for pregnant women residing in specific risk areas [73].

Although Italy is a country at low risk for TBE, the province of Belluno (Veneto), with its rate of notification of cases of TBE of 5.95 per 100,000 population, would be classified by the World Health Organization (WHO) as a highly endemic area (i.e., >five cases per 100,000 population). In conformity with WHO recommendations, the Veneto Regional Authority therefore decided to offer vaccination free of charge to all residents in the province of Belluno, and at a reduced price to residents of other areas of the Veneto region [74].

In Friuli Venezia Giulia, vaccination has been offered free of charge to the whole population since 2013 [75]. The autonomous provinces of Trento and Bolzano have provided vaccination free of charge for all residents and at-risk workers since 2018 [76,77].

Although official data on TBE vaccination rates in Italy are lacking, some reports suggest that vaccination coverage is low, with percentages ranging between 10% and 40%, even in high-risk groups in highly endemic areas [78]. In Friuli Venezia Giulia (FVG), subjects vaccinated until 2016 numbered 60,914, corresponding to a mean coverage of 5%. Among subjects aged 6–16 years, 10.5% received one dose of vaccine, whereas among subjects aged >16 years, 4.6% received vaccination [79].

In order to improve TBE vaccination coverage in the pandemic era, a new approach has recently been adopted. This “drive-through” approach has proved successful in immunizing mountain communities in the province of Belluno during the COVID-19 pandemic. Overall, a total of 12,152 doses of TBE vaccine (first, second, third and booster doses) have been administered to 12,083 people. The “drive-through” approach has proved to be safe and efficient, ensuring continuity of the service during a milder phase of the pandemic. It has also met with users’ approval, despite the few drive-through sites available (in comparison with the more numerous ambulatory facilities) and the traveling distances [80].

## 9. TBE as a Travel Risk

TBE is endemic in some of the most popular holiday destinations in Europe. In 6 of the 10 most visited countries, including Austria and Scandinavia, TBE is endemic in at least some areas.

The real number of travel-associated TBE infections is under-reported for many different reasons, most importantly lack of awareness and under-diagnosing in non-endemic areas.

On assuming that the number of reported TBE cases is proportional to the number of visitors to TBE-endemic areas, the estimated risk of infection is from 1:77,000 to 1:200,000 visits to such areas. In one Austrian study, the risk of TBE infection during a 4-week stay in Austria was calculated to be 1:10,000 [81]. The risk depends on the time of travel (e.g., summer vs. winter), the duration of the stay, and the activities undertaken [27].

Another study evaluated the risk of infection in international travelers to Western/Central Europe, and found that the attack rate could be approximately 0.5–1.3 per 100,000 (1 per 77,000–200,000). The authors concluded that, as advised by the WHO, vaccination should be recommended for travelers exposed to outdoors in rural endemic areas during the period of transmission. Notably, as many at-risk travelers are unaware of the risk, they do not consult competent health professionals; others may consult late, and in some countries of origin, no TBE vaccine is marketed [82].

In recent years, travel-related cases in non-endemic countries have been reported in Israel, the Netherlands, Australia, the United States, and England. While the first autochthonous TBE cases in the Netherlands have recently been reported, most TBE cases diagnosed in the Netherlands are still imported cases involving travelers, mainly from Germany and Austria. In 2019, the first acquired cases of TBE in the United Kingdom were reported in travelers returning from Germany. These examples highlight both the role of endemic holiday areas in the importation of TBE into non-endemic areas and the importance of tracing the travel history of patients with encephalitis, in order to not miss TBE in patients with CNS infection [27].

Although the recommendation to undergo vaccination is particularly strong for people traveling to endemic areas, awareness of the risk of contracting TBE when traveling to such regions is much lower than the perceived risk of contracting hepatitis A or B (about 30 vs. 70%) [83,84]. These findings suggest that the perceived low risk of exposure to TBE among travelers to endemic regions causes these subjects to underestimate the need for pre-travel vaccinations. There is therefore a need to increase travelers’ awareness of the risk of TBE and its prevention, and travel clinics could play an important role in this process [83,84]. In the context of travel medicine, it is important to emphasize that the rapid immunization schedules, with an interval of 2 weeks between the first two doses, have been seen to elicit high antibody titers 14 days after administration of the second dose (89.3%), and 7 days after administration of the third dose (91.7%) [43]. Furthermore, one study [51] observed that vaccine effectiveness was consistently high in subjects who had only received two doses. This information is important for clinicians who provide guidance for travelers to TBE endemic regions, as the median time within which a traveler seeks health advice prior to departure is 3–4 weeks, which is sufficient to administer the first two doses only. These data indicate that individuals traveling to endemic areas are probably protected against TBE for at least 5 months after receipt of the two primary doses of FSME-IMMUN/TICOVAC, which is sufficient for short-term travelers.

## 10. Current Recommendations on TBE Prevention

Vaccination is considered the best means of curbing TBE in endemic regions. This is reflected by the current recommendations issued by international Institutions and Scientific Societies.

Specifically:-The WHO recommends vaccination for the whole population living in areas where TBE is endemic (the average pre-vaccination incidence of clinical disease is ≥5 cases/100,000 population per year) [63]. Inclusion of vaccination against TBE in immunization programs at the regional or national level should be considered. Because the disease tends to be more serious in individuals aged 50–60 years, this age group constitutes an important target for immunization.-The International Scientific Working Group on Tick-Borne Encephalitis (ISW-TBE) recommends vaccination for everyone living in or traveling to TBEV-endemic areas who may be exposed to ticks [85].-Vaccination is also recommended by the European Academy of Neurology (EAN), as recently reported by a consensus review on the prevention, diagnosis, and management of TBE in Europe [16].-The Central European Vaccination Awareness Group (CEVAG) strongly recommends the introduction of universal TBE vaccination in children >1 year of age in countries at very high risk of TBE infections, since vaccination coverage is not yet considered sufficient to control the disease [86].-According to the ECDC recommendations, immunization is recommended for people who live in TBE risk areas or who frequently visit forests and grasslands in TBE risk areas. People who plan to visit TBE risk areas in Europe and Russia should consult their local doctor or immunization services for advice about TBE immunization. Infected dairy animals can shed the TBE virus in their milk, and there are documented cases of TBE transmission through infected milk, which is the one reason why unpasteurized milk or dairy products should not be consumed in TBE risk areas [87].

## 11. Conclusions

As highlighted in this manuscript, TBE is an increasingly worrisome public health problem and can cause severe neurological disease. TBE is endemic in several European countries, and its incidence has recently increased [88]. Various factors may explain this phenomenon, at least in part: social factors (changes in human behavior, duration and type of leisure activities), ecological factors (effects of climate change on the tick population and reservoir animals), and/or technological factors (improved diagnostics, increased medical awareness). Furthermore, a recent study indicated that demographic change (population aging) could have a role in the rising TBE trend.

Importantly, the real burden of TBE is not completely known, as the performance of surveillance systems is suboptimal and cases of disease are under-reported in several areas. For example, in Italy, despite the clinical severity of the disease and the fact that notification is mandatory, under-reporting is still a critical issue [12]. From the public health standpoint, the cases that are overlooked can impair the ability of the National Health Service to conduct epidemiological surveys and take appropriate preventive action [12]. In order to obtain a more realistic epidemiological picture of TBE, it is essential to use multiple data sources and integrate different data-sets with the aim of improving surveillance system. The routine integration of different databases in order to build a single, web-based system in which all actors (physicians, laboratory technicians, public health workers, emergency room staff, etc.) can promptly input all the available information is crucial to implementing targeted prevention strategies [12].

Furthermore, better diagnostic procedures also help to identify possible cases of TBE. In a recent Italian study that used nucleic acid amplification tests (NAATs), TBEV was identified as the etiologic agent of 3.3% of acute aseptic meningitides, encephalitides, and meningoencephalitides (AAMEMs) in adults, suggesting that TBEV should be considered as a possible cause of any AAMEM of unknown origin [89]. Therefore, clinicians should consider the diagnosis of TBE in the patients with a neurological or febrile illness who have recently returned from a TBE-endemic country or reside in endemic areas, particularly if a tick bite or possible tick exposure is reported.

In the last few years, the Italian Ministry of Health has been working to revise the National Surveillance Plan for TBE in an effort to improve notification procedures and minimize under-reporting [42,90]. The main objectives of the Surveillance Plan include: (1) early identification of human cases, in order to adapt public health measures; (2) proper treatment of identified cases; and (3) early prevention and control of possible endemic foci [42,90].

Since we have no means of interrupting the transmission of the virus in nature, vaccination is the mainstay of containment of TBE.

TBE vaccines are effective (protective effect of approximately 95% after completion of the basic vaccination) after three doses and are well tolerated. However, their uptake in endemic areas is suboptimal. The results of a population survey conducted in 2015 indicate that TBE vaccination coverage in Europe is highly variable (from 10% in Finland to 85% in Austria), and that overall vaccination rates are low [91].

The implementation of widespread vaccination programs is essential to containing the disease. Raising awareness of TBE, its consequences, and the benefits of vaccination in both endemic and non-endemic countries, among travelers, professionals working outdoors, and their employers, is extremely important in order to increase vaccination coverage [31]. In addition, the harmful effect of incomplete immunization should be publicized [91].

In conclusion, more attention should be devoted to the economic burden of TBE, in order to guide prevention policies. The cost-effectiveness assessment (CEA) of TBE vaccination should be performed in order to include vaccines in the national vaccination program or to extend current programs. All CEA studies conducted so far have demonstrated the cost-effectiveness of TBE vaccination. However, in many countries, including Italy, such studies are still lacking.

## Figures and Tables

**Figure 1 microorganisms-10-00464-f001:**
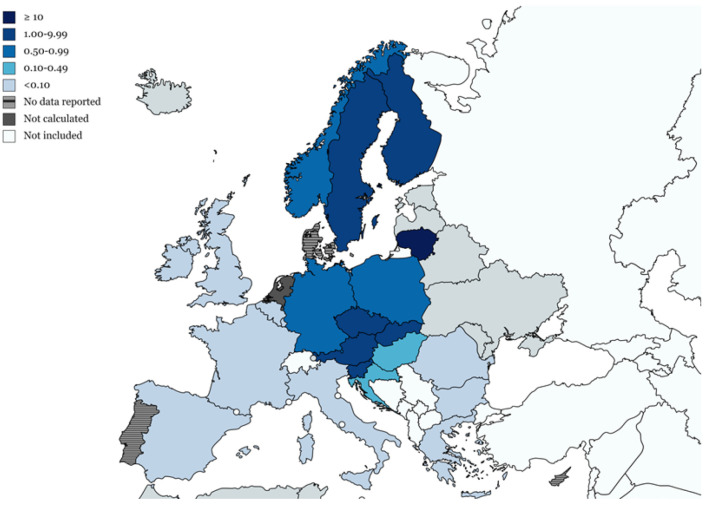
Distribution of confirmed tick-borne encephalitis cases per 100,000 population by country; EU/EEA, 2019 Source: Country reports from Austria, Belgium, Bulgaria, Croatia, Czechia, Estonia, Finland, France, Germany, Greece, Hungary, Ireland, Italy, Latvia, Lithuania, Luxembourg, the Netherlands, Norway, Poland, Romania, Slovakia, Slovenia, Spain, Sweden, and the United Kingdom (Adapted from [13]).

**Figure 5 microorganisms-10-00464-f005:**
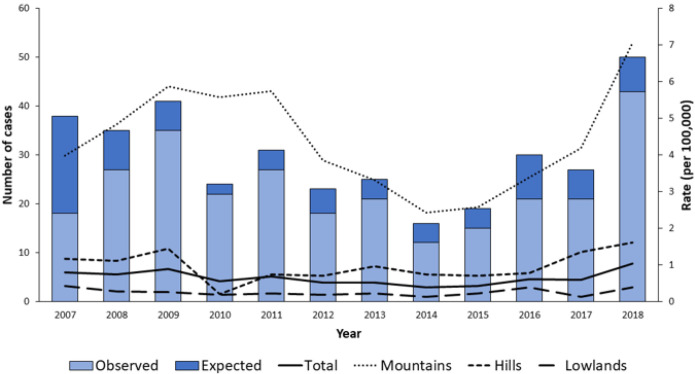
Trend of the adjusted rate and number of cases by geographical area (2007–2018) (Adapted from ref. [12]).

## Data Availability

Not applicable.

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
