# Peer review of "Vaccination against Tick-Borne Encephalitis (TBE) in Italy: Still a Long Way to Go"

_microorganisms, 2022, doi:10.3390/microorganisms10020464_

Round 1

Reviewer 1 Report

This is another review on TBE; there is a focus on Italy where the infection has been neglected in the past. The English is a bit clumsy, the text would benefit from a review by a native speaker. 

MINOR POINTS

38-82: There is a fundamental question whether the introduction should start with a textbook-like summary. Essentially every medical student in endemic areas should know that. So many reviews start with similar boring lines. At least I would suggest the excellent review by Ruzek D et al. Antiviral Res. 2019; 164:23-51.

82: Reference 12 is old, suggest to quote only the latest among the annual ECDC data, your ref. 15.

99: Same, this is an obsolete comparison: We are now 12 years after 2009 and there have been publications showing the recent increase, 2018 and 2020 were record years in many countries. Suggest to delete the sentence 99-101.

124-128: This can be summarized more elegantly into a single sentence.

146: Suggest you consider also the recently published data (Ricco M, Acta Biomed. 2021, Nov 3) in this paragraph.

227/253-262: Economic burden; would suggest to move that to you Chapter #7 starting on line 354.

269-273: It is incorrect to state 'identical', as Neudorf is not the same as K23

297: Since this is a Pfizer sponsored review I suggest that you mention that already 2 doses offer excellent protection for a limited period of time, see another Pfizer sponsored article by Erber W et al. Vaccine. 2021: in press. "Effectiveness of TBE vaccination in Southern Germany and Latvia."

351: Suggest to add reference Schmidt AJ et al. J Travel Med. 2021.

417: Suggest to mention the fascinating action reported by De Polo et al. in J Press Med Heg. 2020; 61:E497-500.

444: There are far more recent data on risk in travelers as compared to [74], see e.g. Steffen R in J Travel Med. 2016;taw018 or Hills S in J Travel Med. 2021;taab167.

472: Be more specific, mention the threshold set by WHO

Author Response

Reviewer 1

This is another review on TBE; there is a focus on Italy where the infection has been neglected in the past.

The English is a bit clumsy, the text would benefit from a review by a native speaker.

Reply: The manuscript has been revised by native speaker.

38-82: There is a fundamental question whether the introduction should start with a textbook-like summary. Essentially every medical student in endemic areas should know that. So many reviews start with similar boring lines. At least I would suggest the excellent review by Ruzek D et al. Antiviral Res. 2019; 164:23-51.

Reply: Thank you for this comment. We have now completely revised the introduction section, by contextualizing it to the manuscript aim. The suggested article has been consulted (Ruzek D et al. Antiviral Res. 2019; 164:23-51).

82: Reference 12 is old, suggest to quote only the latest among the annual ECDC data, your ref. 15.

Reply: Done

99: Same, this is an obsolete comparison: We are now 12 years after 2009 and there have been publications showing the recent increase, 2018 and 2020 were record years in many countries. Suggest to delete the sentence 99-101.

Reply: As required we have deleted the sentence 99-101 and we have updated the epidemiological data.

124-128: This can be summarized more elegantly into a single sentence.

Reply: Done.

146: Suggest you consider also the recently published data (Ricco M, Acta Biomed. 2021, Nov 3) in this paragraph.

Reply: Done.

227/253-262: Economic burden; would suggest to move that to you Chapter #7 starting on line 354.

Reply: Done.

269-273: It is incorrect to state 'identical', as Neudorf is not the same as K23

Reply: We agree with reviewer comment, and we corrected the text accordingly.

Reviewer 2 Report

Comments to: Manuscript ID: microorganisms-1541300, Title: Vaccination against tick-borne encephalitis (TBE) in Italy: still a long way to go.

General comments.

The manuscript provides a review of European TBE epidemiology, vaccination recommendations ect. with emphasis on the current state of affairs in Italy. The review is comprehensive, well written and provide much detailed information. I do not often see papers at this scientific standard. I do however find it difficult to identify a clear take-home message. With this, I have but a few comments, which follows below.

Major comments.

  1. It is essential that a map of Italy is provided because non-Italian readers will be unable to follow the many references to Italian geography/provinces and municipals.
  2. It is stated that “Thanks to the mass vaccination program, TBE is still under control in Austria, with 50-100 cases per year and more than 4000 cases prevented between 2000 and 2011 [55]” - which might have been an acceptable statement in 2015, but as the ECDC numbers in reference [15], shows – number are increasing in recent years.
  3. There are inconsistencies and/or divergences in the assessment of acceptable booster intervals and the impact of age, which could be more clearly communicated.

Ad 1. Please add a map, which specifies the relevant locations that you are referring to. I suspect that you can leave out Southern Italy, so that the details of the Northern provinces are clearly identifiable. I spent a great deal of time googling the locations, but gave up on assessing the spatial consistency, and are thus unable to assess whether your claim of underreporting makes sense – from that particular perspective. I ended up wondering whether there were cases in Lombardia – along the border to Switzerland. I would suspect that there would be cases there, just as there is at the Austrian border.   Please bear in mind that there is a great deal of focus on the changes in the epidemiology at the edges of the TBE-range – so accuracy will be important for epidemiologists.

If you want to complete the map on page 3, for Northern EU countries the following may be useful.  For TBE in Denmark: https://www.ssi.dk/sygdomme-beredskab-og-forskning/sygdomsleksikon/t/tbe (it is simple text so you may use google translate). It just underlines that the shift in TBE towards the north-west also apply here. Recently Ixodes ricinus appeared as far north as Iceland https://pubmed.ncbi.nlm.nih.gov/29017579/ - no TBE yet.

Ad 2. The number of cases in Austria in 2018 was 170. Also, the health authorities reported about 25 cases of death due to TBE (WHO mortality database, ICD A84.1) over the past decade – which with a CFR of 1.5% indicate a CFR-projected incidence of 166 per year. So stating that TBE is “under control” can be debated.

The authors of the paper: Rubel, F., Walter, M., Vogelgesang, J. R., & Brugger, K. (2020). Tick-borne encephalitis (TBE) cases are not random: explaining trend, low-and high-frequency oscillations based on the Austrian TBE time series. BMC infectious diseases, 20(1), 1-12. – did much to explain/dismiss this in their analysis. I am not sure that I accept that a 10% increase in population size explains a doubling of the case number – but I will leave it to you to asses possible explanations. I’m fine as long as you acknowledge that something has changed – leading to an increase in numbers.

Ad. 3. The text relating to vaccine strategies, their efficacy and the booster-shot interval, needs tidying up a bit, which perhaps is a question of writing more actively. You are correctly reporting various findings: Line 305 (5 y), line 312 and 346 (10 y), but refrain from taking ownership of the totality: Please clarify what you believe is the most prudent interpretation.

The same applies to the effect of age. You refer to declining antibodies in Line 309: a reference to the mentioned study is missing – and later in Line 338 : your refer to [54] in stating that vaccines are effective regardless of age.  The missing reference may be: Hainz, U., Jenewein, B., Asch, E., Pfeiffer, K. P., Berger, P., & Grubeck-Loebenstein, B. (2005). Insufficient protection for healthy elderly adults by tetanus and TBE vaccines. Vaccine23(25), 3232-3235.  – which states that aging is a problem.  Please clarify what you believe is the most prudent interpretation.

Combining 2 and 3. I accept that it may be difficult to include the above in the current text, without disrupting the flow.  The solution may simply be to include a discussion of the “consensus”. The paper leaves me with a suspicion that the “way to go in Italy” is riddled with challenges. It seems that:

  • TBE vaccines are effective and safe – but most likely less effective in the elderly and in people with underlying co-morbidities (the latter is typically excluded in studies which creates a gap between the clinical evidence and the epidemiological record)
  • The European population is growing – primarily because people live longer (size and age is confounded). This is not in itself an issue or a cause for major epidemiological shifts. The problem is that this aging population continues to be active at a much higher age than the previous generations. You may here emphasize the rapidly aging of the Italian population (as compared to Scandinavians).
  • It could be proposed that:
    • In also including the insights gained from the COVID-19 vaccination campaigns, that shortened booster intervals should be considered for the elderly and for people with co-morbidities/inflammation disorders.
    • We are at risk of falsely concluding that TBE vaccination is less effective (in an aging Austrian population) – when we perhaps should argue that it is more important than ever before.
  • Which lead towards the conclusion that vaccination is an essential tool for minimizing the public health impact of TBE, but it is not a magic want, which remains unaffected by environmental and demographic change. Such limitations should not be used to dismiss the importance of vaccination.
  • It is sadly an impossible task to explain your success, when this is given by an “absence (of disease)”!

I am NOT saying that you need to buy into this line of argumentation – I am simply saying that the paper would be much improved, if you seek to connect the evidence and deliver a frame of understanding, such as possible at this particular time. It may, in this, be relevant to point out important gaps in our knowledge. Ironically, this need is the direct product of the thoroughness and details provided. I have no doubt that the given authors can meet this challenge.

In avoiding that the paper gets too long I recommend that you delete paragraphs and sections, which are less relevant. This include:

  • The entire section 9, because travel medicine, to me, is a separate issue.
  • Line 221 to 226, as you just repeat a generic explanation.
  • Line 406-407, as you already stated this.

Last, I suggest that you condense the paragraphs such that each is at least 10 lines of text, because the 3-4 line paragraphs are challenging the reading flow in this particular layout.

Minor comments

Line 23: delete “of TBE”

Line 63: make a blank new line to separate the text and let the two previous paragraphs merger with the upper text.

Line 101: delete the last bit “including Russia” – its understood.

Line 217: If 3 dies, then the CFR-projected incidence is: 3/0.015 = 200 !

Line 316: Missing reference – ELISA prone to cross-reactions. Somehow, communicate that this is not the case in diagnostics due to varying cross reactivity.  

Line 414: “less endemic” – rephrase.

Line 439 – if you keep this section, then please add a few more references.

Section 11. The paper does not need a lengthy summary. Rewrite this section and present your interpretation such as it appear with the current evidence. (see above ad. 2 and 3).  Note, in achieving clarity – “less is more”, and bullet-points are often quite useful. Perhaps the title of this should connect back to the title: “the road ahead” or to emphasize “la via italiana avanti”

Fig.1 Its impossible to read the legend – increase the size.

Fig 2. With my eyesight – I need larger numbers on the x and y axis.

Fig. 5. The x-axis text legend is a bit “fluffy” – may require that you save the figure at higher resolution or that you add text-boxes with “an overwrite”.

Thank you for giving me the opportunity to read this paper…

Author Response

Reviewer 2

The manuscript provides a review of European TBE epidemiology, vaccination recommendations with emphasis on the current state of affairs in Italy. The review is comprehensive, well written and provide much detailed information. I do not often see papers at this scientific standard. I do however find it difficult to identify a clear take-home message. With this, I have but a few comments, which follows below.

Major comments

  1. It is essential that a map of Italy is provided because non-Italian readers will be unable to follow the many references to Italian geography/provinces and municipals.

Reply: The map of the Italy showing areas where TBE is present has been inserted (figure 2).

  1. It is stated that “Thanks to the mass vaccination program, TBE is still under control in Austria, with 50-100 cases per year and more than 4000 cases prevented between 2000 and 2011 [55]” - which might have been an acceptable statement in 2015, but as the ECDC numbers in reference [15], shows – number are increasing in recent years.

Reply: We have now updated the section “Epidemiology of TBE in Europe” analyzing in detail the Austrian data.

  1. There are inconsistencies and/or divergences in the assessment of acceptable booster intervals and the impact of age, which could be more clearly communicated.

Reply: As required, the assessment of booster intervals and the impact of age have been considered in the section 6 (Focus on FSME-IMMUN/TICOVAC vaccine).

Ad 1. Please add a map, which specifies the relevant locations that you are referring to. I suspect that you can leave out Southern Italy, so that the details of the Northern provinces are clearly identifiable. I spent a great deal of time googling the locations, but gave up on assessing the spatial consistency, and are thus unable to assess whether your claim of underreporting makes sense – from that particular perspective. I ended up wondering whether there were cases in Lombardia – along the border to Switzerland. I would suspect that there would be cases there, just as there is at the Austrian border. Please bear in mind that there is a great deal of focus on the changes in the epidemiology at the edges of the TBE-range – so accuracy will be important for epidemiologists.

Reply: As suggested, we have added the map of Italy showing areas where TBE is present. Furthermore, we have researched studies relative to Lombardy Region. On the basis of the Italian official surveillance system, no notifications of TBE cases were registered in Lombardy region (region of northern Italy bordered by Switzerland) despite the vector Ixodes ricinus is present.

If you want to complete the map on page 3, for Northern EU countries the following may be useful. For TBE in Denmark: https://www.ssi.dk/sygdomme-beredskab-og- forskning/sygdomsleksikon/t/tbe (it is simple text so you may use google translate). It just underlines that the shift in TBE towards the north-west also apply here. Recently Ixodes ricinus appeared as far north as Iceland https://pubmed.ncbi.nlm.nih.gov/29017579/ - no TBE yet.

Reply: We have now updated the section “Epidemiology of TBE in Europe” as requested by reviewer 1, however we did not modified figure 1.

Ad 2. The number of cases in Austria in 2018 was 170. Also, the health authorities reported about 25 cases of death due to TBE (WHO mortality database, ICD A84.1) over the past decade – which with a CFR of 1.5% indicate a CFR-projected incidence of 166 per year. So stating that TBE is “under control” can be debated.

The authors of the paper: Rubel, F., Walter, M., Vogelgesang, J. R., & Brugger, K. (2020). Tick- borne encephalitis (TBE) cases are not random: explaining trend, low-and high-frequency oscillations based on the Austrian TBE time series. BMC infectious diseases, 20(1), 1-12. – did much to explain/dismiss this in their analysis. I am not sure that I accept that a 10% increase in population size explains a doubling of the case number – but I will leave it to you to asses possible explanations. I’m fine as long as you acknowledge that something has changed – leading to an increase in numbers.

Reply: The manuscript of Rubel et al. has been considered and the comment of their results has been reported in section 2 “Epidemiology of TBE in Europe”.

Combining 2 and 3. I accept that it may be difficult to include the above in the current text, without disrupting the flow. The solution may simply be to include a discussion of the “consensus”. The paper leaves me with a suspicion that the “way to go in Italy” is riddled with challenges. It seems that:

TBE vaccines are effective and safe – but most likely less effective in the elderly and in people with underlying co-morbidities (the latter is typically excluded in studies which creates a gap between the clinical evidence and the epidemiological record)

The European population is growing – primarily because people live longer (size and age is confounded). This is not in itself an issue or a cause for major epidemiological shifts. The problem is that this aging population continues to be active at a much higher age than the previous generations. You may here emphasize the rapidly aging of the Italian population (as compared to Scandinavians).

It could be proposed that:

In also including the insights gained from the COVID-19 vaccination campaigns, that shortened booster intervals should be considered for the elderly and for people with co- morbidities/inflammation disorders.

We are at risk of falsely concluding that TBE vaccination is less effective (in an aging Austrian population) – when we perhaps should argue that it is more important than ever before.

Which lead towards the conclusion that vaccination is an essential tool for minimizing the public health impact of TBE, but it is not a magic want, which remains unaffected by environmental and demographic change. Such limitations should not be used to dismiss the importance of vaccination.

It is sadly an impossible task to explain your success, when this is given by an “absence (of disease)”!

I am NOT saying that you need to buy into this line of argumentation – I am simply saying that the paper would be much improved, if you seek to connect the evidence and deliver a frame of understanding, such as possible at this particular time. It may, in this, be relevant to point out important gaps in our knowledge. Ironically, this need is the direct product of the thoroughness and details provided. I have no doubt that the given authors can meet this challenge.

Reply: Thank you for this comment. We have now completely revised the manuscript considering all comments of the reviewers.

In avoiding that the paper gets too long I recommend that you delete paragraphs and sections, which are less relevant. This include:

The entire section 9, because travel medicine, to me, is a separate issue.

Reply: As the reviewer 1 suggested to improve the section 9 considering other studies, we have decided to maintain this section.

Line 221 to 226, as you just repeat a generic explanation.

Reply: Done.

Line 406-407, as you already stated this.

Reply: Done.

Last, I suggest that you condense the paragraphs such that each is at least 10 lines of text, because the 3-4 line paragraphs are challenging the reading flow in this particular layout.

Reply: Done.

Minor comments

Line 23: delete “of TBE”

Reply: Done.

Line 63: make a blank new line to separate the text and let the two previous paragraphs merger with the upper text.

Reply: As suggested by Reviewer 1, we have completely revised the introduction section.

Line 101: delete the last bit “including Russia” – its understood.

Reply: As suggested by Reviewer 1, the sentence has been deleted.

Line 217: If 3 dies, then the CFR-projected incidence is: 3/0.015 = 200 !

Reply: The sentence has been deleted.

Line 316: Missing reference – ELISA prone to cross-reactions. Somehow, communicate that this is not the case in diagnostics due to varying cross reactivity.

Reply: The paragraph 6 has been rewritten. We included also new data of FSME-IMMUN/TICOVAC vaccine.

Line 414: “less endemic” – rephrase.

Reply: As required, the new sentence is “In conformity with WHO recommendations, the Veneto Regional Authority therefore decided to offer vaccination free of charge to all residents in the Province of Belluno, and at a reduced price to residents of other areas of the Veneto Region”.

Line 439 – if you keep this section, then please add a few more references.

Reply: We have decided to keep paragraph 9 and, as suggested by reviewer 1, we have added new references.

Section 11. The paper does not need a lengthy summary. Rewrite this section and present your interpretation such as it appear with the current evidence. (see above ad. 2 and 3). Note, in achieving clarity – “less is more”, and bullet-points are often quite useful. Perhaps the title of this should connect back to the title: “the road ahead” or to emphasize “la via italiana avanti”

Reply: Thank you for this comment. We have now revised the conclusions section focusing the key points of the manuscript.

Fig.1 Its impossible to read the legend – increase the size.

Reply: Done.

Fig 2. With my eyesight – I need larger numbers on the x and y axis.

Reply: Done.

Fig. 5. The x-axis text legend is a bit “fluffy” – may require that you save the figure at higher resolution or that you add text-boxes with “an overwrite”.

Reply: Figure 5 has been deleted.

Round 2

Reviewer 1 Report

Overall well revised, just minor details to be considered:

29  "... vaccination included ... in program" is not specific. Do you mean universal recommendation (like Austria) or just recommendation for all in endemic areas or just for risk groups (with/without financial coverage)?

123  Be more precise: EU countries, does not include e.g. Russia, Switzerland

310  Case fatality rate only 0.9% in more recent review by Kohlmaier B et al.  Microorganisms. 2021; 9(7):1420.

361  Update the info: FSME-IMMUN/Ticovac for several years is now marketed by Pfizer, Encepur had been sold many years ago to GSK and recently from there to Bavarian Nordic. You corrected that only 368-372.

380  Not precise, suggest to mention minimum interval between 2nd and 3rd dose which is more relevant. 

403  There are now a better references than the SmPC (which is impossible to click on with your limited indication): Pugh S et al. J Travel Med. 2022 Jan 6; taab193. Erber W et al. Vaccine. 2022 Jan 31; 40(5):819-25. You mention that only in 451 but that should be moved up.

497  This is contested by recent Schmitt HJ et al. Vaccines (Basel). 2021 Aug 21; 9(8):932.

696  Empty line

Reference 59 = 68

Author Response

Reviewer 1

Overall well revised, just minor details to be considered:

29  "... vaccination included ... in program" is not specific. Do you mean universal recommendation (like Austria) or just recommendation for all in endemic areas or just for risk groups (with/without financial coverage)?

Reply: In abstract, one clarification has been inserted.

123 Be more precise: EU countries, does not include e.g. Russia, Switzerland

Reply: On clarification has been inserted in “Epidemiology of TBE in Europe” session.

310  Case fatality rate only 0.9% in more recent review by Kohlmaier B et al.  Microorganisms. 2021; 9(7):1420.

Reply: As required, in “Clinical burden of TBE” session we have inserted the data from the manuscript by Kohlmaier et al. Microorganism 2021.

361 Update the info: FSME-IMMUN/Ticovac for several years is now marketed by Pfizer, Encepur had been sold many years ago to GSK and recently from there to Bavarian Nordic. You corrected that only 368-372.

Reply: Done

380  Not precise, suggest to mention minimum interval between 2nd and 3rd dose which is more relevant.

Reply: Done

403  There are now a better references than the SmPC (which is impossible to click on with your limited indication): Pugh S et al. J Travel Med. 2022 Jan 6; taab193. Erber W et al. Vaccine. 2022 Jan 31; 40(5):819-25. You mention that only in 451 but that should be moved up.

Reply: Done

497  This is contested by recent Schmitt HJ et al. Vaccines (Basel). 2021 Aug 21; 9(8):932.

Reply: The reference (Schmitt HJ et al. Vaccines (Basel). 2021 Aug 21; 9(8):932) has been cited in the correct point of the text.

696  Empty line

Reply: Done

Reference 59 = 68

Reply: Thank you, we have corrected the references.

Reviewer 2 Report

Dear editor and authors,

Thank you for giving me the opportunity to read this excellent paper. I note that the authors did a good job in their revision and I have therefore only a few comments. The comments addresses a few additional terminological and linguistic issues.

Line: 18. I think you need to decide whether you mean Climate Change (the concept) or shifting weather patterns. "climate changes" is neither of these.

Line 24: I don’t like the phrase “etiological treatment” – perhaps “antiviral therapy” is better.

Line 25: the virus does not have a life-cycle (it lacks stages) – replace with transmission.

Line 26: the word containment is used for epidemics of human-to-human transmitted diseases. For zoonotic diseases, the terms are “prevention and control”.

Line 36: replace “extremely important” with “critical”

Line 91:   replace “time” with “period”

Line 96: as in line 24.

Line 97: write: “increasing public health concern”

Line 99: replace “as” with “and”

Line 106: replace “sustainable” with “viable”

Line 154: replace “issued” with “published” or “acknowledged”

Line 159: delete “entire”

Line 167: replace “value” with “number”

Line 272: replace “ had not been vaccinated” with “was unvaccinated”

Line 311: replace “ not negligible” with “considerable”

Line 404: start with “Consistently,”

Line 445: fix the indent

Line 566: replace “vaccination against TBE” with “TBE vaccination”.

Line 633: delete “as”

Line 708: as in line 97.

Line 753: as in line 25.

Line 762: as in line 26.

For further assessment of differences in TBE awareness in Italy. Find the Google Trends home-page. Type tick borne encephalitis (pick the term with the subscript “Disease” – and assess the map of Italy.

Author Response

Reviewer 2

Dear editor and authors,

Thank you for giving me the opportunity to read this excellent paper. I note that the authors did a good job in their revision, and I have therefore only a few comments. The comments addresses a few additional terminological and linguistic issues.

Line: 18. I think you need to decide whether you mean Climate Change (the concept) or shifting weather patterns. "climate changes" is neither of these.

Reply: “Climate Changes” has been amended with “Climate change”.

Line 24: I don’t like the phrase “etiological treatment” – perhaps “antiviral therapy” is better.

Reply: Done

Line 25: the virus does not have a life-cycle (it lacks stages) – replace with transmission.

Reply: Done

Line 26: the word containment is used for epidemics of human-to-human transmitted diseases. For zoonotic diseases, the terms are “prevention and control”.

Reply: Done

Line 36: replace “extremely important” with “critical”

Reply: Done

Line 91:   replace “time” with “period”

Reply: Done

Line 96: as in line 24.

Reply: Done

Line 97: write: “increasing public health concern”

Reply: Done

Line 99: replace “as” with “and”

Reply: Thank you for your suggestion, but we refer to maintain “as”.

Line 106: replace “sustainable” with “viable”

Reply: Done

Line 154: replace “issued” with “published” or “acknowledged”

Reply: Done

Line 159: delete “entire”

Reply: Done

Line 167: replace “value” with “number”

Reply: Done

Line 272: replace “had not been vaccinated” with “was unvaccinated”

Reply: Done